# Feasibility, usability and acceptability of paediatric lung ultrasound among healthcare providers and caregivers for the diagnosis of childhood pneumonia in resource-constrained settings: a qualitative study

Atif Riaz,[1] Olga Cambaco,[2] Laura Elizabeth Ellington [iD],[3] Jennifer L Lenahan,[4] Khatia Munguambe,[5] Usma Mehmood,[1] Alessandro Lamorte,[6] Sana Qaisar,[1] Benazir Baloch,[1] Neel Kanth,[7] Muhammad Imran Nisar,[1] Giovanni Volpicelli,[8] Quique Bassat,[9,10] Fyezah Jehan [iD],[1] Amy Sarah Ginsburg [iD] [11]

**Correspondence to**
Laura Elizabeth Ellington; lelling@uw.edu

## ABSTRACT

**Objectives** Paediatric pneumonia burden and mortality are highest in low-income and middle-income countries (LMIC). Paediatric lung ultrasound (LUS) has emerged as a promising diagnostic tool for pneumonia in LMIC. Despite a growing evidence base for LUS use in paediatric pneumonia diagnosis, little is known about its potential for successful implementation in LMIC. Our objectives were to evaluate the feasibility, usability and acceptability of LUS in the diagnosis of paediatric pneumonia.

**Design** Prospective qualitative study using semistructured interviews

**Setting** Two referral hospitals in Mozambique and Pakistan

**Participants** A total of 21 healthcare providers (HCPs) and 20 caregivers were enrolled.

**Results** HCPs highlighted themes of limited resource availability for the feasibility of LUS implementation, including perceived high cost of equipment, maintenance demands, time constraints and limited trained staff. HCPs emphasised the importance of policymaker support and caregiver acceptance for long-term success. HCP perspectives of usability highlighted ease of use and integration into existing workflow. HCPs and caregivers had positive attitudes towards LUS with few exceptions. Both HCPs and caregivers emphasised the potential for rapid, improved diagnosis of paediatric respiratory conditions using LUS.

**Conclusions** This was the first study to evaluate HCP and caregiver perspectives of paediatric LUS through qualitative analysis. Critical components impacting feasibility, usability and acceptability of LUS for paediatric pneumonia diagnosis in LMIC were identified for initial deployment. Future research should explore LUS sustainability, with a particular focus on quality control, device maintenance and functionality and adoption of the new technology within the health system. This study highlights the need to engage both users and recipients of new technology early in order to adapt

## Strengths and limitations of this study

► This is the first study to evaluate important elements of successful lung ultrasound (LUS) implementation through healthcare providers and caregiver perspectives.

► We focused on themes of LUS feasibility, usability and acceptability to help individuals and organisations to develop deployment strategies that focus on successful long-term sustainability.

► We performed interviews prior to LUS implementation, requesting participants to reflect on their opinions with limited experience of LUS.

future interventions to the local context for successful implementation.

**Trial registration number** NCT03187067.

## INTRODUCTION

Pneumonia kills over 900 000 children under 5 years of age worldwide each year, the vast majority in low-income and middle-income countries (LMIC).[1 2] Paediatric pneumonia is challenging to diagnose, especially in settings where clinical expertise and diagnostic tools are not readily available or accessible. Chest radiography (CXR) is expensive, resource-intensive, carries ionising radiation and lacks sensitivity and specificity.[3–5] In LMIC, healthcare providers (HCPs) rely on nonspecific signs and symptoms, including cough, difficulty breathing, fast breathing and chest indrawing, for the diagnosis and management of pneumonia based on the WHO guidelines for Integrated Management of Childhood

Illness (IMCI).[6] Paediatric lung ultrasound (LUS) has emerged as a promising diagnostic tool for pneumonia in LMIC due to its diagnostic accuracy and reliability, portability, ease of use, lack of ionising radiation and lower cost compared with CXR.[7–15] Despite a growing evidence base for LUS use in paediatric pneumonia diagnosis, particularly in high-income settings, little is known about its potential for successful implementation in LMIC.

The research-to-practice gap is a well-described challenge, with evidence-based interventions taking an average of 17 years to be integrated into clinical practice, and even longer in LMIC.[16] This gap concedes that evidence alone for an intervention is not sufficient to ensure its successful uptake and sustainability in real-world settings. Implementation in LMIC carries additional challenges related to resource availability, clinical training, maintenance of devices and time constraints.[17–19] Successful implementation strategies suggest first assessing potential barriers and facilitators in order to adapt the intervention to the local context.[20] Therefore, the objective of this study was to evaluate the feasibility, usability and acceptability of LUS through qualitative methods with HCPs and caregivers in Mozambique and Pakistan.

## METHODS

We performed a qualitative evaluation as part of a larger pilot study evaluating whether adding LUS to WHO IMCI clinical assessment improves pneumonia diagnosis in young children. Briefly, 270 children aged 2–23 months presenting with cough and/or difficulty breathing were enrolled through routine screening at the two study sites in Mozambique and Pakistan. Children underwent LUS and CXR in addition to routine clinical assessment and were followed for a month after enrollment. Diagnostic accuracy of LUS was compared with CXR as well as to WHO IMCI clinical assessment alone. The complete protocol has been presented elsewhere.[21]

### Setting

We recruited participants at two sites, Manhiça District Hospital (MDH) in Manhiça, Mozambique and Sindh Government Children's Hospital-Poverty Eradication Initiative (SGCH-PEI) in Karachi, Pakistan. Located 90 km from the capital city Maputo in a semirural district, MDH is a referral healthcare facility serving a population of 183 000. MDH has a 32-bed paediatric ward, 8-bed high-dependency unit and 6-bed paediatric day hospital, and is typically staffed with at least 2 paediatricians, 6 general physicians and larger numbers of nurses and medical agents. Under-five mortality rate in Mozambique is estimated to be 87.2/1000 live births with 19% of postneonatal deaths (aged 1–59 months) caused by pneumonia.[22] SGCH-PEI is the district hospital for District Central, the largest district in Karachi, Pakistan. SGCH-PEI is equipped with more than 100 inpatient paediatric beds including a neonatal intensive care unit and has approximately 1000–1500 paediatric outpatient visits per day.

SGCH-PEI is staffed with four paediatricians and a larger number of nurses. Under-five mortality rate in Pakistan is estimated to be 85.5/1000 live births with pneumonia the number one cause of postneonatal deaths, at 29%.[23] Neither site has access to subspecialty paediatric consultation. Digital CXR is available at both sites.

### Participants

Participants were recruited from within the larger pilot study between April and May 2018 and included both HCPs and caregivers. Healthcare administrators and HCPs were eligible to participate if all of the following criteria were met: (1) employed at one of the study sites; (2) involved in or aware of the LUS study and (3) had experience caring for children. HCPs were enrolled by convenience sampling, targeting a total of 10 at each site to balance healthcare administration duties, experience with LUS and HCP role. Caregivers were eligible if they were at least 18 years of age and were a primary caregiver for a child enrolled in the study. Caregivers were approached sequentially about participating in this study during their child's enrolment in the larger pilot study until the target enrolment of ten caregivers at each site was reached.

### Data collection

Researchers JLL and ASG developed the in-depth interview guides with review by AR and KM. Using the standardised interview guides and standard operating procedures to reduce interinterviewer variability within and between sites, AR and KM, experienced in qualitative methods, supervised the onsite trainings of qualitative research assistants from the local community. Semistructured interviews were conducted in person in the participant's preferred language by the trained research assistants using the standardised interview guide that did not change throughout data collection (online supplemental S1). Interviews lasted 30–45 min in duration and were performed in a private space within each healthcare setting. For HCPs and administrators, questions explored feasibility and acceptability of LUS, while HCPs were also asked specifically about usability of LUS. Feasibility questions explored challenges with healthcare delivery in its current state, experience with technology, storage of equipment and additional perceptions of facilitators and barriers to integration of LUS within their healthcare facility. Usability questions targeted user's experience with LUS, particular difficulties with the device and preferred level of HCPs to perform LUS. Acceptability questions focused on likes and dislikes regarding LUS, perceived value of LUS and comparison to CXR. Questions for caregivers centred around acceptability of LUS in the care of their child, including caregivers' understanding of LUS, their likes and dislikes with the LUS examination, comparison to their experience with CXR for their child and preference for availability of LUS in the future. All interviews were digitally recorded, transcribed with deidentified information and translated

into English. Transcriptions were reviewed and edited for grammar and clarity and therefore, were not considered direct quotes.

## Data analysis

Participant characteristics were summarised using descriptive statistics (counts and proportions). Qualitative thematic analyses were performed by OC and KM in Mozambique, AR in Pakistan and LEE using deductive coding. All had previous experience in qualitative coding and analysis. A deductive approach was used by manually coding transcripts using a priori topical codes chosen based on the primary research aims of this study: evaluation of feasibility, usability and acceptability. We were less interested in developing new themes by an inductive approach and instead chose to employ an existing framework using themes describing implementation. *Feasibility* was defined as how easily HCPs/administrators thought LUS could be integrated into the existing healthcare setting and included both individual and organisational attributes around perceived facilitators and barriers to using LUS within routine workflow. *Usability* was defined as the extent to which LUS operation was user-friendly in obtaining images and interpreting them. *Acceptability* was defined as the extent to which individuals performing (HCPs) or receiving (caregivers) the LUS examination considered it to be appropriate. These definitions were adapted from the implementation science literature.[24] Data analysis was supported by NVivo V.11.0 software in Pakistan, while in Mozambique, a matrix was manually developed using Microsoft Excel. Final factors and chosen quotes were reviewed by all authors across sites to ensure agreement of key findings. SRQR checklist for qualitative research was used to guide reporting of our methods and results.[25]

## Patient and public involvement

Patients were not involved in the design, or conduct, or reporting or dissemination plans of this research.

## RESULTS
## Participant characteristics

A total of 21 HCPs and 20 caregivers were enrolled from both sites. Of the 21 HCPs, 9 were physicians, 3 of whom had primarily administrative roles, 4 were nurses or a medical agent, 2 were radiology technicians and 6 had other primary roles (table 1). The LUS examinations were performed by non-physician healthcare personnel at both sites, a nurse and a medical agent in Mozambique and two radiology technicians in Pakistan. Over half of the HCPs who participated in the study were female (55% in Pakistan and 60% in Mozambique). Eight HCPs were physicians in Pakistan (73%), while only one HCP was a physician in Mozambique (10%). Similarly, eight (73%) had graduated from university or higher in Pakistan, while two (20%) had in Mozambique. About half of the HCPs at both sites had more than 8 years of experience in

**Table 1** Characteristics of healthcare provider participants

| | Mozambique n=10 | Pakistan n=11 |
|---|---|---|
| | N (%) | |
| Female | 6 (60) | 6 (55) |
| Age (years) | | |
| 20–35 | 4 (40) | 3 (27) |
| 36–60 | 6 (60) | 8 (73) |
| Provider role | | |
| Physician | 1 (10) | 8 (73) |
| Healthcare administrator* | 0 | 3 |
| Nurse or medical agent | 4 (40) | 0 |
| Radiology technician | 0 | 2 (18) |
| Research manager | 1 (10) | 0 |
| Phlebotomist | 0 | 1 (9) |
| Consent administrator | 4 (40) | 0 |
| Highest education achieved | | |
| Primary school | 2 (20) | 0 |
| Secondary school | 6 (60) | 1 (9) |
| Technical school | 0 | 2 (18) |
| University or higher | 2 (20) | 8 (73) |
| Years of experience in current role | | |
| 1–8 | 5 (50) | 6 (55) |
| 9–14 | 3 (30) | 3 (27) |
| >14 | 2 (20) | 2 (18) |

*Healthcare administrators were also physicians.

their current role. Caregivers were all female and ranged in age from 19 to 42 years (table 2). All but one caregiver was the mother of a child enrolled in the parent study, and all but one had a child aged less than 12 months enrolled. The majority of caregivers had completed primary (60%) or secondary (30%) school. Interviews with HCPs and caregivers revealed major subthemes around feasibility, usability and acceptability of LUS implementation for diagnosing and managing childhood pneumonia (table 3).

## Feasibility

HCPs identified three major factors affecting LUS feasibility: costs/resources; support from policymakers and acceptance of caregivers. Identified costs associated with successful LUS implementation included equipment, appropriately trained staff and time. About half of the HCPs interviewed in both Mozambique and Pakistan expressed concern with the cost of the LUS technology as a potential barrier for the widespread implementation of LUS as a diagnostic modality. Currently, ultrasound is not a readily available diagnostic tool in their settings and would require the purchase of a number of devices for use. HCPs in each site had different concerns about safe storage of the device. For example, one HCP in Pakistan

| Table 2 | Characteristics of caregiver participants | |
|---|---|---|
| | Mozambique n=10 | Pakistan n=10 |
| | N (%) | |
| Female | 10 (100) | 10 (100) |
| Age (years) | | |
| 19–24 | 4 (40) | 3 (30) |
| 25–29 | 0 | 3 (30) |
| 30–34 | 3 (30) | 2 (20) |
| ≥35 | 3 (30) | 2 (20) |
| Age of child (months) | | |
| <12 | 10 (100) | 9 (90) |
| 12–23 | 0 | 1 (9) |
| Relationship to child | | |
| Parent | 9 (90) | 9 (90) |
| Grandparent | 1 (10) | 0 |
| Other | 0 | 1 (10) |
| Highest education achieved | | |
| Less than primary school | 0 | 1 (10) |
| Primary school | 9 (90) | 3 (30) |
| Secondary school | 1 (10) | 4 (40) |
| University or higher | 0 | 2 (20) |

and a majority in Mozambique were concerned about potential theft. A minority of HCPs, mostly in Mozambique, also expressed concern about maintenance of the LUS equipment over time.

> …I bet it's not that affordable because if it was affordable, even the district hospital could have it since it is a large unit, and it would be something helpful to have.
>
> HCP, Mozambique

> It is portable, small and delicate device so there are chances of theft. And since we are working in public sector hospital, so we have to be very careful for its protection.
>
> HCP, Pakistan

> With poor maintenance, the quality of the functioning of the device will be compromised.
>
> HCP, Mozambique

In addition to the equipment itself, HCPs also commented on the importance of recruiting and training staff to perform LUS. About half of the HCPs in both Mozambique and Pakistan quoted that successful LUS implementation relied on adequately trained staff to operate and interpret LUS.

> Train more staff… to do LUS and interpret. I think the important thing is to train … first, to train people. So, to be able to implement, then, the next phase, we will need serious lectures to help explain to the

people… what is ultrasound, what is its purpose, and what the benefits of ultrasound are. Also, of course, one training session isn't enough because the hospital is very large.

> HCP, Mozambique

> They have to increase the number of technicians, staff, and other requirements so that everything will be done in an organized way.
>
> HCP, Pakistan

Training for the pilot study consisted of a 1-day standardised LUS training course as well as 3 days of closely supervised LUS practice at each site. Opinions on adequate length of training varied greatly across the two sites. In Pakistan, one HCP thought 2–3 days would suffice, while the remaining three who responded suggested at least 2 weeks would be required for adequate training. Notably, no participants mentioned refresher or repeat training for successful implementation.

Performing LUS may take time away from other provision of care in LMIC, as one HCP in Mozambique remarked:

> Currently, when a CXR is ordered, the patient leaves while we are seeing other patients or in other activities. Then the patient comes back with the CXR. In the case of ultrasound, it has to be the clinician who is attending, doing the ultrasound, interpreting the ultrasound, and giving the result. So maybe it would take a little more time.
>
> HCP, Mozambique

Similarly, HCPs discussed the impact of LUS on time for the care of young children with respiratory illnesses. The majority of HCPs felt that LUS was fast and efficient due to its small size and portability. However, a few also felt that in a busy clinical setting, even if LUS is quick, it may be challenging to perform because of time restraints related to other essential duties.

A second important factor affecting feasibility of LUS was support from policymakers. This was brought up by HCPs with and without administrative roles in Pakistan. The majority of HCPs in Pakistan identified the importance of key stakeholder buy-in to promote policy change on a broader scale to improve implementation and sustainability of LUS in the public hospital sector.

> An ultrasound and probe are so costly, and we have to purchase them. We would require documentation for purchase, and this process could be delayed. It depends on administration. They are the decision makers, and they can help us.
>
> HCP, Pakistan

Interestingly, this was not a theme that HCPs in Mozambique identified for successful LUS implementation; rather HCPs in Mozambique agreed that policymakers would gladly receive LUS, but only one HCP commented on the importance of policymakers for successful LUS

**Table 3** Summary of major subthemes

| Domain | Subthemes | Explanation | Example questions from interview guide |
|---|---|---|---|
| Feasibility | Cost/resources | Includes cost and maintenance/storage of LUS equipment, sufficient number of adequately trained staff, potential increased workload and time | What should happen to make LUS successful? Do you foresee any problems with keeping the LUS device in the hospital when not in use? Do you think that LUS could be integrated into this facility? What would be some of the challenges/ barriers? How much training (hours and/or days) do you think that learning to use LUS would take? |
| | Support from policymakers | Allocating sufficient resources to LUS equipment purchase and promoting use of LUS broadly across public hospital sector | |
| | Acceptance of caregivers | Buy-in from recipients of LUS | |
| Usability | Device management | Performing LUS image/video clip capture | Tell me about your experience with ultrasound. What was easy and/or difficult to learn regarding LUS? How much time did a LUS examination take? What level of healthcare provider do you think should perform/ interpret LUS? |
| | Image interpretation | Making diagnosis using available LUS images/video clips | |
| | Integration into existing workflow | Using LUS as point-of-care tool within existing clinic structure | |
| Acceptability | Perceived value | Reasons respondent gave for liking and/or wanting LUS implemented within healthcare setting | What did you think about LUS? What do you like about LUS? What do you dislike? Did you have any concerns about LUS during your child's visit? How do you think that the LUS would impact your child's care (caregivers) or ability to care for children (HCPs)? How does your experience with LUS compare to CXR? |
| | Affective attitude | Extent respondent liked or did not like LUS | |
| | Patient comfort | How child and caregiver tolerated LUS examination | |
| | General understanding | Respondent's ability to articulate indications and expectations of LUS | |

CXR, chest radiography; LUS, lung ultrasound.

implementation. Of note, no healthcare administrators were interviewed in Mozambique.

A third factor impacting feasibility of LUS was acceptance among caregivers. HCPs reported that for successful use, LUS must be accepted by caregivers through increased knowledge and practice. Most HCPs felt that caregivers would have high acceptability of LUS, but all highlighted the importance of caregiver buy-in for LUS to be successful. One HCP also mentioned the importance of acceptance by HCPs through rigorous research.

It is necessary to first explain to the children's mothers. Mothers still do not know and do not understand exactly what an ultrasound is, so it is necessary for the provider to explain to the mother in order to make them realize that this is a very important means of diagnosis. That, and that it doesn't hurt.

HCP, Mozambique

Difficulties may be whether people are accepting it or not and what people think of it.

HCP, Pakistan

Advocacy is best way to introduce [LUS], and dissemination of information is also a good way to create an awareness among people. First, you have to disseminate it to healthcare providers and give them

confidence that it has good sensitivity, its diagnoses are correct, and you can save children from x-ray's radiations. If they are convinced, they promote it to the patients.

HCP, Pakistan

## Usability

HCPs with experience using the LUS device found the probe and interface easy to use, especially after practice, and that capturing LUS images was more straightforward than interpretation. Generally, HCPs felt that LUS was easiest to use and interpret in calm children without abnormal findings.

This application is very good. We performed it very well. We didn't face any difficulty in it. Everything is good in it. Sometimes little pathology in x-ray is not visible, but these are visible in this [LUS]… in my opinion it is good. This device is good. It is comfortable. Its probe is also good. So I didn't face any difficulty with it.

HCP, Pakistan

As said, ultrasound is simple. It's simply a matter of capturing images… The most difficult thing is to

understand the meaning of the images and to interpret. This is the most difficult part.

HCP, Mozambique

Almost half of HCPs quoted challenges with high quality image capture when children were irritable or crying or if the identified pathology was small. A minority also reported difficulties with the LUS system, particularly with storing or losing images that were recently captured. Almost all HCPs agreed that HCPs who performed LUS could interpret the results.

Well, I think that who knows how to take the exam has to know how to interpret.

HCP, Mozambique

HCPs at the two sites differed in their opinions regarding what level of HCPs should perform LUS examinations. In Pakistan where two radiology technicians performed the LUS examinations, HCPs generally agreed that radiologists or sonographers would be the most appropriate HCPs to perform this task compared with physicians, nurses or other technicians who were evaluating the child. In Mozambique where a nurse and a medical agent performed the LUS examinations, the majority of HCPs agreed that a nurse and/or physician with or without a technician could perform the LUS examination.

Interpretation can be done by a senior level person who must know all about ultrasound. Like, sonologists and physicians can do it.

HCP, Pakistan

… for me, it does not have to be a doctor to perform it and know how to interpret it. So I think a health technician, a nurse, a medical agent can do it very well.

HCP, Mozambique

Another aspect of LUS usability involved its ready integration within existing workflows. All HCPs agreed that LUS could be integrated within the existing workflow due to its small size, relatively quick examination and portability. HCPs reported that LUS took about 15–20 minutes to complete.

This is portable device and easy to carry and handle…. The visibility of [the screen] is good and its quality is also good and a clear [video] clip can be saved with it.

HCP, Pakistan

As a whole, I think it is easy to handle. It is accessible and easy to transport. You can leave it here, for example, to an area that is without electricity, and bring it to do the exam room and make a diagnosis. I think this is the most impressive of ultrasound.

HCP, Mozambique

## Acceptability

HCPs and caregivers reported four major factors affecting acceptability of the LUS device: perceived value of LUS;

affective attitude; patient comfort and general understanding of LUS. All HCPs and caregivers expressed that they liked LUS overall and described the perceived value of LUS, highlighting the rapid diagnostic ability of LUS, which included guiding definitive treatment during initial consultation instead of 'guessing' the diagnosis.

Yes, it [should be incorporated into routine care], because as we have always had this problem of [bad] breathing in children. I think it would improve or help a lot. Now only those who enter the [research] study are lucky.

HCP, Mozambique

I was satisfied because they do this to see the child's health, to know if the child is in good health or not, the child's breathing, here in the child's ribs and heart.

Caregiver, Mozambique

It can be useful for doctor to immediately diagnose what has happened to child; otherwise they give medicine by guessing disease. I liked this ultrasound… It helps diagnose the disease quickly and it saves from wrong treatment due to accurate diagnosis.

Caregiver, Pakistan

Caregivers were specifically asked if they would prefer bringing their child to a facility with LUS compared with one without, and all said yes.

HCPs also described benefits of LUS as a potential alternative compared with CXR with the primary and most common benefit reported being the lack of ionising radiation with LUS. Just under half of the HCPs emphasised the additional benefit of streamlining workflow by using LUS as a point-of-care tool at the child's bedside, rather than having the child and family travel to the radiology department for a CXR. A few HCPs also added that CXR machines in their healthcare centres were not always functional due to lack of electricity or malfunctioning equipment. All but one caregiver also reported that they liked LUS as much or more than CXR.

We get an instant diagnosis without radiation.

HCP, Pakistan

It could change the dynamics of the unit itself. Because suppose there is a huge queue of patients who want to have an X-ray. They will stay in the queue for a long time. Now, with the pulmonary ultrasound, if there are two, three devices, there is an examiner over there, another one over there and ready, the child entered, examined, and left. That would streamline this in a way.

HCP, Mozambique

While the vast majority of comments regarding LUS were positive, a few caregivers from Pakistan commented that their children were agitated during the LUS examination. Two caregivers were worried that their children's agitation was an indication that LUS was harmful to their

children. No caregivers from Mozambique reported concerns with LUS.

> Ultrasound takes more time [than CXR], and the child also gets irritated.
>
> Caregiver, Pakistan
>
> I was worried that my child should not get hurt. I was afraid a little bit.
>
> Caregiver, Pakistan

When asked about their understanding of LUS, all HCPs were able to correctly identify the indication for LUS. A minority of caregivers were still unclear about the purpose of LUS after the examination was explained and performed by study staff. Some HCPs and caregivers had higher expectations of LUS than what was explained to them.

> It shows clear picture of inside body part of child tells about all hidden diseases.
>
> Caregiver, Pakistan
>
> I like it because the probe makes it easier for me to diagnose problems, such as pneumonia and other pathologies that I can see when I have an ultrasound scan.
>
> HCP, Mozambique

## DISCUSSION

There is a growing body of literature supporting the use of LUS for the diagnosis of paediatric pneumonia across many healthcare settings. However, few studies have looked at feasibility of LUS deployment in LMIC, and none have explored perspectives of those using or receiving the intervention: HCPs and caregivers. This study therefore adds to the implementation science literature of new technologies in LMIC for improved diagnosis and management of acute respiratory illness in young children. This was the first study to evaluate HCP and caregiver perspectives of paediatric LUS through qualitative analysis, specifically their thoughts on feasibility, usability and acceptability of LUS in their healthcare setting. Themes associated with LUS feasibility from the HCP perspective included concerns of working in a resource-constrained environment, namely cost of equipment, limited trained staff and time constraints, as well as the importance of policymaker support and caregiver acceptance. Regarding LUS usability, HCPs highlighted ease of image capture compared with interpretation, challenge of performing LUS in small, crying children, appropriate experience level of HCP and integration into existing workflow. LUS acceptability themes from both HCPs and caregivers emphasised perceived value of LUS, such as rapid diagnosis, no ionising radiation compared with CXR and streamlining workflow, as well as the importance of caregiver/HCP understanding of LUS. Among both HCP and caregivers, there was an overall positive attitude with few exceptions.

## Quality control

In this study, HCPs had varying opinions regarding length of a LUS training programme, from a few days to multiple weeks. While high quality initial training is critically important to ensure successful roll-out of an intervention, perhaps more important is developing a system for quality monitoring over time with opportunities for refresher training. For example, as part of our pilot study, we monitored quality of LUS acquisition and interpretation in real-time. One of the study sites had lower quality images identified early in data collection. In response to this, we performed a series of remote refresher trainings via video conferencing. We tailored this intervention to the needs of the end user, first again reviewing basic use of the LUS probe for high quality image capture and followed up with a session on using the LUS probe in different ways to investigate pathology. Quality control is also important to address the high turnover of HCPs and the overwhelming shortage of highly skilled HCPs that require task-shifting to less specialised HCPs.[26–29] These shortages underscore the need for ongoing quality control monitoring and efficient, high quality and readily available training. Fortunately, there are different options to address these challenges in LMIC, including development of local experts to train the trainers, growing opportunities for remote learning through video conferencing and reducing the need for training by incorporating artificial intelligence into LUS image acquisition and interpretation.

## Device maintenance and functionality

While the importance of maintenance and equipment functionality was not a major theme identified by the HCPs or administrators in this study, these are important factors to consider in the adoption of LUS (or any new technology). The ultrasound system used in this study consisted of three parts: a probe, a removable cord and a third-party tablet with an application as the collection interface. Various parts of this system failed at different points during the study and required replacement parts shipped from the USA. To make LUS adoption feasible long-term, the ultrasound selected should have a local supplier able to provide maintenance and replacements, as well as technical documentation and support in the local language. Second, prior to scale-up, budgets for devices and projections for device maintenance should be clearly defined by organisations providing and receiving the intervention. Third, provisions for back-up parts and devices should be explored with an understanding of who is responsible for repairing the device and what is an acceptable turn-around time for repair.

Our findings were similar to prior qualitative research on the implementation of point-of-care ultrasound and more broadly, point-of-care diagnostics in LMIC.[30–32] Key stakeholder perspectives revealed that training as well as cost and maintenance of materials were major barriers to use, while improved diagnostic accuracy, timely diagnosis and portability were major benefits.[30–32] When

considering point-of-care diagnostics more broadly, stakeholders emphasised cost-effectiveness evaluations and improving quality management systems, again supporting the importance of cost containment and quality of healthcare delivery through trained staff and functional equipment for sustainability.[31]

## Acceptance and adoption of new technology

Quality control and device functionality ensure that the building blocks for successful implementation are in place and can be sustained over time. However, these building blocks are meaningless without the buy-in from healthcare leadership and frontline providers. Our study supports high acceptability of LUS by both HCPs and caregivers. While acceptability is an important first step, it is not sufficient to ensure high uptake by health systems. The Consolidated Framework for Implementation Research highlights the importance of evaluating the inner setting (eg, structural characteristics, culture and teamwork) and outer setting (eg, external policies and incentives) to determine successful implementation.[33] This means that the setting in which the intervention is applied is almost as important as the intervention itself. For LUS to be successful, engagement by policymakers to develop guidelines, policy statements and incentives will encourage systematic uptake by health centres. Furthermore, working with HCPs to integrate LUS into existing workflows within their busy clinical setting should attempt to streamline patient care without adding to the already high burden of care.

## Limitations

There were several limitations to this study. While we performed training with interviewers prior to data collection to increase consistency across sites, the differences between sites may have contributed to interviewers asking questions differently, leading to differing responses between sites. Although we interviewed a wide range of participants with different experiences, it is possible that we missed important perceptions limited by our sample size. We were unable to recruit healthcare administrators in Mozambique. Therefore, HCP responses in Mozambique reflected the opinions of frontline HCPs and not stakeholders in healthcare administration as was the case in Pakistan. We also recruited HCPs with varying experience with LUS, some of whom were involved in the larger pilot study. Participation by some interviewees in the LUS pilot study may have contributed to response bias, with HCPs responding more favourably to LUS and neglecting to discuss their negative opinions. Additionally, although transcripts were transcribed and translated, it is possible that some important concepts were missed or misinterpreted in the analysis process.

Importantly, HCPs discussed themes that were most relevant to initial deployment of LUS in their healthcare setting. This study was performed as part of a pilot study prior to implementation in their healthcare facility. HCP experience was therefore limited to brief exposure to the device without experiencing first-hand challenges with sustainability. Future directions should include understanding barriers and facilitators to sustainability in LMIC following real-world experience with a particular focus on quality control, device maintenance and functionality and acceptance and adoption of the new technology.

## CONCLUSIONS

Through qualitative analysis, we identified several important components impacting feasibility, usability and acceptability of LUS for the diagnosis of paediatric pneumonia in LMIC. HCPs and caregivers liked LUS for its perceived rapid results and the potential for improved diagnostic accuracy of pneumonia at bedside, lack of ionising radiation and potential for improved clinic workflow. HCPs thought it could be successfully integrated into their healthcare setting with sufficient training, knowledge sharing, policymaker buy-in and caregiver acceptance. Potential barriers included cost and maintenance of LUS equipment, adequately trained staff and comfort level of both HCPs and caregivers with a new technology. Taken together, this study highlights the importance of early engagement of both users and recipients of new technology in order to adapt future interventions to the local context for successful implementation.

**Author affiliations**
[1]Pediatrics and Child Health, Aga Khan University, Karachi, Pakistan
[2]Centro de Investigação em Saúde de Manhiça, Manhica, Maputo, Mozambique
[3]Pediatrics, University of Washington School of Medicine, Seattle, Washington, USA
[4]Save the Children Federation, Seattle, Washington, USA
[5]Centro de Investigacao em Saude de Manhica, Manhica, Maputo, Mozambique
[6]Emergency Medicine, Umberto Parini Hospital, Aosta, Valle d'Aosta, Italy
[7]Children's Hospital-Poverty Eradication Initiative, Sindh Government Hospital Karachi, Karachi, Sindh, Pakistan
[8]Emergency Medicine, San Luigi Gonzaga University Hospital, Orbassano, Italy
[9]ISGLOBAL, Barcelona, Spain
[10]Universitat de Barcelona, Barcelona, Catalunya, Spain
[11]Clinical Trial Center, University of Washington, Seattle, Washington, USA

**Acknowledgements** We thank the dedicated study staff at Manhiça District Hospital in Manhiça, Mozambique and Sindh Government Children's Hospital–Poverty Eradication Initiative in Karachi, Pakistan for implementing the study and providing patient care. We thank Cayetana Verastegui and Carla Pinto in Manhiça, Mozambique for their contributions to the study. We also thank the trial participants, their caregivers and the local community in Manhiça, Mozambique and Karachi, Pakistan for their participation and support. CISM is supported by the Government of Mozambique and the Spanish Agency for International Development. ISGlobal receives support from the Spanish Ministry of Science and Innovation through the 'Centro de Excelencia Severo Ochoa 2019-2023' Program (CEX2018-000806-S), and support from the Generalitat de Catalunya through the CERCA Program.

**Contributors** JLL and ASG designed the study and data collection instruments with input from AR, KM, AL, MIN, GV, QB and FJ. AL and GV provided lung ultrasound training and support. JLL and ASG coordinated and supervised data collection from the sites and NK, MIN, FJ and QB supervised their respective sites. AR, OC, KM, UM, SQ, BB and MIN acquired and managed the data. AR, OC and KM analysed and interpreted the data and produced study reports. LEE analysed and interpreted the data using information from the study reports and wrote the manuscript with critical input from JLL and ASG. All authors worked collaboratively to review and approve the final manuscript.

**Funding** This work was supported by grants from the Bill and Melinda Gates Foundation (OPP1105080) and Save the Children.

**Competing interests** None declared.

**Patient consent for publication** Not required.

**Ethics approval** The study was conducted in accordance with the International Conference on Harmonisation, Good Clinical Practice and the Declaration of Helsinki 2008, and was approved by the Western Institutional Review Board in the state of Washington, USA; the Comité Institucional de Bioética em Saúde do Centro de Investigação em Saúde de Manhiça (Manhiça, Mozambique, Ref. 084/2017); the Comité Nacional de Bioética em Saúde (Maputo, Mozambique, Ref. 246/CNBS/17); the Comité de Ética del Hospital Clínic de Barcelona (Barcelona, Spain; Ref. HCB/2017/0074) and the Aga Khan University Ethics Review Committee (Karachi, Pakistan). All participants provided written informed consent in their preferred local language. Participant anonymity was maintained during data analysis and report writing.

**Provenance and peer review** Not commissioned; externally peer reviewed.

**Data availability statement** Data are available upon reasonable request. Data will be made available upon reasonable request to the corresponding author.

**ORCID iDs**
Laura Elizabeth Ellington http://orcid.org/0000-0001-7904-4249
Fyezah Jehan http://orcid.org/0000-0002-5874-4358
Amy Sarah Ginsburg http://orcid.org/0000-0002-2291-2276

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
