## [Reviewer comments · BMJ Open]

ARTICLE DETAILS

TITLE (PROVISIONAL)	Feasibility, usability, and acceptability of pediatric lung ultrasound among healthcare providers and caregivers for the diagnosis of childhood pneumonia in resource-constrained settings: A qualitative study
AUTHORS	Riaz, Atif; Cambaco, Olga; Ellington, Laura; Lenahan, Jennifer; Munguambe, Khatia; Mehmood, Usma; Lamorte, Alessandro; Qaisar, Sana; Baloch, Benazir; Kanth, Neel; Nisar, Muhammad; Volpicelli, Giovanni; Bassat, Quique; Jehan, Fyezah; Ginsburg, Amy Sarah

VERSION 1 – REVIEW

REVIEWER	Reza Rasti Karolinska Institutet, Department of Global Public Health, Stockholm, Sweden
REVIEW RETURNED	26-Aug-2020

GENERAL COMMENTS	I applaud you for including a qualitative study like the one you present, in the implementation process of your larger LUS project in Pakistan and Mozambique. There are numerous examples of various medical technologies being deployed to LMICs without a priori efforts to understand barriers towards their intended use. Your report is interesting and well written. I have some minor comments before recommending acceptance of your paper; 1. please provide include the interview guide that was designed and used for the interviews.2. It would be interesting to see an overview of the coding structure that was performed during data analysis.3. page 5 line 16: it is in the nature of qualitative research to not be representative of other possible settings, there is no need for this to be excused as a limitation4. page 6 line 41: 'Implementation in LMIC...constraints.' - do you have a reference for this statement?5. page 7 'Setting' section: how are the two hospitals staffed? What is the burden of pneumonia in these regions/countries? What is their overall childhood mortality?6. page 7 'Participants' section where all 3 criteria to be met for HCPs, or any one of them?7. page 9 line 33: 'Content analysis' is a separate method for qualitative data analysis. I suggest to rephrase, perhaps into 'Data analysis was supported by...'8. page 9 'Ethics' section: why the need to include ethics approval from Spain and the US? Also, what is the reference number for approval from the first stated ethics review board in Mozambique?
--

	9. page 10 'Ethics': where the study participants compensated in any way? 10. page 10: Your clinicaltrials.gov registration does not include this qualitative study. I suppose it was registered for your larger LUS project. If not applicable to this study, then remove from this manuscript, or rephrase so that it is understood that the larger project was registered. 11. page 10 line 48: one caregiver was not the parent of a patient; however, the caregiver criteria given before stated that caregivers were eligible if atleast 18 years and had a child participating in the LUS study. 12. Regarding quotes: even if tempting, it is generally advised not to identify the different respondents when presenting quotes. No need to identify them as HCP X or HCP Y. HCP Mozambique would be sufficient. 13. Page 14 line 12: 'increasing' to be rephrased to 'recruiting'?
--	---

REVIEWER	Steve Graham University of Melbourne, Australia
REVIEW RETURNED	31-Aug-2020

GENERAL COMMENTS	Many thanks for the opportunity to review this original work, reporting positive and/or readily amenable responses to the introduction of a new technology with practical potential for LMIC settings. Has integration to improve clinical evaluation such as accurate measurement of respiratory rate been considered? It might be helpful for the reader to see a picture of a device/probe in use on an infant for example.
---

REVIEWER	Elizabeth M Molyneux College of Medicine, Malawi /UK
REVIEW RETURNED	08-Jan-2021

GENERAL COMMENTS	In this paper the authors report the findings of a qualitative study, undertaken in 2 referral paediatric units, of the opinions of health care providers and caregivers of using lung ultrasound (LUS) in place of chest xrays (CXR). Semi-structured interviews that focussed on feasibility, usability and acceptability were undertaken. The hospitals are very different; one hospital in Manhica, Mozambique is a district referral hospital supported by an international research unit. It has 32 paediatric, 8 high dependency and 6 day-beds. The hospital in Pakistan has 1000-12000 paediatric outpatients a day, 100 inpatient beds and an intensive care unit. The hospital in Mozambique is staffed mainly by clinical officers (termed medical 'agents' by the authors) and nurses. In Pakistan the hospital has specialist staff including radiologists and sonographers. This is a sub-study within a mother study of the impact on diagnostic accuracy of adding LUS to the clinical signs and symptoms of an Integrated Management of Childhood Illness (IMCI) diagnosis of pneumonia. In the larger study children with a cough or difficulty breathing had both a CXR and LUS carried out. The enrolled children were 2 – 23 months of age. The ultrasounds were carried
--

	out by nurses or a medical agent in Mozambique and sonographers in Pakistan. The number of health care providers (HCP) and caregivers (all but one a mother) recruited to the sub-study was small; 10 in each category at each hospital. The health care providers were further divided into administrative or clinical staff and in Mozambique because there was no administrator, the study consenters and research manager were included instead. The administrative staff were asked especially about the feasibility of replacing CXRs with LUS; the clinical staff were asked about the usability and to some extent feasibility and acceptability; the mothers focused on acceptability. The researchers provide quotations of several of the responses to their questions. When discussing feasibility, the problems of device cost, maintenance and storage were raised. Also, that ultrasonography would only be feasible if acceptable to HCPs and especially, to care givers. Some HCPs in Mozambique felt that as LUS had to be done by the HCPs themselves it would slow them down. For CXRs patients are sent to the xray department and the HCP could continue seeing other patients. It was felt that having 2 or 3 ultrasound machines, (and therefore 2 or 3 sonographers to do the scans) would speed up clinic work. One big advantage was felt to be that the ultrasound was portable, runs on batteries and is not reliant on a permanent power source. HCPs thought that LUS was easy to use but less easy to interpret. In Mozambique staff were satisfied with the one-day LUS training and 3 days supervision received prior to starting the study; interestingly in Pakistan the sonographers felt that a 2 week training course was needed. No one spoke of quality checks or regular re-training. Surprisingly the staff did not mention, though the authors do, the problem of rapid turnover of ward and clinic staff and the constant need to train new staff members. The caregivers were all aged 19-42 years and most were satisfied with the LUS. When explained to them, they understood its use. One mother thought it took too long. All the mothers, except one, had babies aged <12 months; i.e., of an age when it may be difficult to keep a restless, fast breathing baby still enough for the ultrasound to be good enough to interpret. It is hard to generalise from this small study with such differences between the two hospital sites, numbers and staff. It would be useful to know how many CXRs are taken daily, on average, not in a study but in routine care. We do not know about the quality of
--	--

interpretations of the LUS (or indeed the CXRs) which are somewhat subjective. We are not told how the diagnosis of pneumonia was made, i.e., what the sonographer was asked to look for, nor whether the same probe was used for all sizes and age of children. In other studies, the most commonly identified ultrasound findings include pulmonary consolidation, positive air bronchogram, abnormal pleural line, and pleural effusion¹. The portability, low running costs and ease of use make it a very attractive diagnostic tool for poorly resourced centres. Several studies have shown LUS to be accurate in diagnosing paediatric pneumonia: a child's relatively thin chest wall, unossified ribs and small lung volume all make the procedure easier to undertake than in an adult¹. Also, the fact that the baby can be held on the mother's lap, sitting upright, breast feeding or lying supine are reassuring to both mother and child, making the procedure easier to carry out. Pereda et al in a meta-analysis of 5 studies, in which LUS were compared to CXRs, found that LUS had a sensitivity of 96% (95% confidence interval [CI]: 94%-97%) and specificity of 93% (95% CI: 90%-96%), and positive and negative likelihood ratios were 15.3 (95% CI: 6.6-35.3) and 0.06 (95% CI: 0.03-0.11), respectively². But the ultrasounds were done and interpreted by experienced, trained sonographers. Tsou PY et al report a significant difference ($p < 0.001$) between the accuracy of novice versus experienced sonographers³.

The cost of ultrasound machines compares very well with those of an xray unit; running costs are minimal and as long as the probe is not dropped, maintenance and replacement costs are not onerous. Experience suggests that users be limited to a few people so that the device is not mishandled. Most decisions about the acquisition of diagnostic devices must take into account who will be able to use them, and here is the rub – if the department is small there may not be sufficient use of ultrasound to keep a user proficient, or if there is a high turnover of staff the amount of training required for new staff will be onerous. The ultrasound can be used in the investigation of many other illnesses, how will the device be retained for clinic use when it is in demand elsewhere? Who will do quality checks of scans and their interpretation?

How many machines are needed for what size of health centre and what is the best probe to buy that will be useful to the most children with different problems?

Will the use of CXR diminish or will LUS be added to what is done at present?

This study was done in 2018, it would be useful to return to the two health units and find out what is being done now? Was the use of ultrasound sustained and if not, why not? If LUS are still being performed are quality checks carried out, what do the stored pictures look like?

	In conclusion this study raises more questions than it answers. The replacement of CXRs with LUSs is theoretically feasible and acceptable but the decision to do this must be context specific and constantly monitored. References  1. Kharasch S, Duggan NM, Cohen AR, Shokoohi H. Lung Ultrasound in Children with Respiratory Tract Infections: Viral, Bacterial or COVID-19? A Narrative Review. Open Access Emerg Med. 2020;12: 275–285. Published online 2020 Oct 14. doi: 10.2147/OAEM.S238702 eCollection (accessed Jan 7th 2021) 2. Pereda MA, Chavez MA, Hooper-Miele CC, Gilman RH, Steinhoff MC, Ellington LE, Gross M, Price C, Tielsch JM, Checkley W. Lung ultrasound for the diagnosis of pneumonia in children: a meta-analysis. Pediatrics. 2015;135:714-22. doi: 10.1542/peds.2014-2833. Epub 2015 Mar 16. (accessed Jan 6th 2021) 3. Tsou PY, Chen KP, Wang YH, Fishe J, Gillon J, Lee CC, Deanehan JK, Kuo PL, Yu DTY. Diagnostic Accuracy of Lung Ultrasound Performed by Novice Versus Advanced Sonographers for Pneumonia in Children: A Systematic Review and Meta-analysis. Acad Emerg Med. 2019;26:1074-1088. (accessed Jan 7th 2021)
--	---

VERSION 1 – AUTHOR RESPONSE

Reviewer: 1

Dr. reza Rasti, Karolinska Institute

Comments to the Author:

Dear authors,

I applaud you for including a qualitative study like the one you present, in the implementation process of your larger LUS project in Pakistan and Mozambique.

There are numerous examples of various medical technologies being deployed to LMICs without a priori efforts to understand barriers towards their intended use.

Your report is interesting and well written.

I have some minor comments before recommending acceptance of your paper;

1. please provide include the interview guide that was designed and used for the interviews.

Response: As requested, we have attached the in-depth interview guides.

2. It would be interesting to see an overview of the coding structure that was performed during data analysis.

Response: As requested, we have shared our coding structure below for your interest. However, we feel that this level of granularity is less appropriate for the manuscript and because of this, included Table 3 to give readers an overview of the themes and subthemes.

Parent codes are left justified. Child codes indented.

Child codes emerged from reviewing the transcripts.

Feasibility

cost

equipment use

maintenance of equipment

time

portability

personnel

organizational/systems/policy

caregivers acceptance

Acceptability

patient comfort

diagnostic accuracy

perceived value

understanding LUS role/concept

comparison to CXR

dissatisfaction

satisfaction (general)

Usability

ease of use/learning

time commitment

training

integration into clinical practice

Barrier

Facilitator

Low-resource setting

HCP

Caregiver

3. page 5 line 16: it is in the nature of qualitative research to not be representative of other possible settings, there is no need for this to be excused as a limitation

Response: Thank you for your comment. As suggested, we have removed this point from the Strengths and Limitations section.

4. page 6 line 41: 'Implementation in LMIC...constraints.' - do you have a reference for this statement?

Response: As suggested, we have added 3 references (17, 18, 19) related to implementation science challenges/ differences in LMIC.

5. page 7 'Setting' section: how are the two hospitals staffed? What is the burden of pneumonia in these regions/countries? What is their overall childhood mortality?

Response: As requested, we have included additional information in the "Setting" section about both study sites regarding hospital staffing, burden of acute lower respiratory infections/pneumonia, and under-five childhood mortality, and burden of acute lower respiratory infections/pneumonia.

6. page 7 'Participants' section where all 3 criteria to be met for HCPs, or any one of them?

Response: All 3 criteria were met for HCP eligibility. We have edited the manuscript to clarify this point.

7. page 9 line 33: 'Content analysis' is a separate method for qualitative data analysis. I suggest to rephrase, perhaps into 'Data analysis was supported by...'?

Response: Thank you for this astute observation. The manuscript has been edited per your recommendation.

8. page 9 'Ethics' section: why the need to include ethics approval from Spain and the US? Also, what is the reference number for approval from the first stated ethics review board in Mozambique?

Response: Study co-investigators were from the USA and Spain and therefore, required ethics approvals from their home institutions as well as the study site institutions. Also, we have included the requested reference number from the ethics review board in Mozambique.

9. page 10 'Ethics': where the study participants compensated in any way?

Response: There was no compensation provided to study participants for participation.

10. page 10: Your clinicaltrials.gov registration does not include this qualitative study. I suppose it was registered for your larger LUS project. If not applicable to this study, then remove from this manuscript, or rephrase so that it is understood that the larger project was registered.

Response: The qualitative component was part of the overall study from its conception, was described in our published protocol, and so was included as part of the clinicaltrials.gov registration.

11. page 10 line 48: one caregiver was not the parent of a patient; however, the caregiver criteria given before stated that caregivers were eligible if atleast 18 years and had a child participating in the LUS study.

Response: Thank you for pointing this discrepancy out. As suggested, we have clarified the eligibility criteria to include caregivers at least 18 years of age and who were a primary caregiver for a child in the study.

12. Regarding quotes: even if tempting, it is generally advised not to identify the different respondents when presenting quotes. No need to identify them as HCP X or HCP Y. HCP Mozambique would be sufficient.

Response: As suggested, we have edited individual respondents out of the quotes.

13. Page 14 line 12: 'increasing' to be rephrased to 'recruiting'?

Response: As suggested, we have made this edit to the manuscript.

Reviewer: 2

Dr. Stephen Graham, Murdoch Childrens Research Institute

Comments to the Author:

Many thanks for the opportunity to review this original work, reporting positive and/or readily amenable responses to the introduction of a new technology with practical potential for LMIC settings. Has integration to improve clinical evaluation such as accurate measurement of respiratory rate been considered? It might be helpful for the reader to see a picture of a device/probe in use on an infant for example.

Response: Thank you for your thought-provoking question. Integration of respiratory rate measurement was beyond the scope of this project, but there is ongoing work by several groups evaluating multimodal measurement technologies for pneumonia diagnosis and management.

Unfortunately, we are not able to include a photo of the device/probe in use on a child.

Reviewer: 3

Prof. Elizabeth Molyneux, Queen Elizabeth Central Hospital, Blantyre

Comments to the Author:

please see review

It is hard to generalise from this small study with such differences between the two hospital sites, numbers and staff.

Response: We acknowledge that generalizability is a limitation of most qualitative work and certainly of this small pilot study. In our paper, we tried to stress the importance of thoughtful assessments of barriers and facilitators prior to introduction of a new technology like LUS in routine practice.

It would be useful to know how many CXRs are taken daily, on average, not in a study but in routine care. We do not know about the quality of interpretations of the LUS (or indeed the CXRs) which are somewhat subjective. We are not told how the diagnosis of pneumonia was made, i.e., what the sonographer was asked to look for, nor whether the same probe was used for all sizes and age of children. In other studies, the most commonly identified ultrasound findings include pulmonary consolidation, positive air bronchogram, abnormal pleural line, and pleural effusion .

The portability, low running costs and ease of use make it a very attractive diagnostic tool for poorly resourced centres. Several studies have shown LUS to be accurate in diagnosing paediatric pneumonia: a child's relatively thin chest wall, unossified ribs and small lung volume all make the procedure easier to undertake than in an adult . Also, the fact that the baby can be held on the mother's lap, sitting upright, breast feeding or lying supine are reassuring to both mother and child, making the procedure easier to carry out. Pereda et al in a meta-analysis of 5 studies, in which LUS were compared to CXRs, found that LUS had a sensitivity of 96% (95% confidence interval [CI]: 94%-97%) and specificity of 93% (95% CI: 90%-96%), and positive and negative likelihood ratios were 15.3 (95% CI: 6.6-35.3) and 0.06 (95% CI: 0.03-0.11), respectively . But the ultrasounds were done and interpreted by experienced, trained sonographers. Tsou PY et al report a significant difference ($p < 0.001$) between the accuracy of novice versus experienced sonographers .

The cost of ultrasound machines compares very well with those of an xray unit; running costs are minimal and as long as the probe is not dropped, maintenance and replacement costs are not onerous. Experience suggests that users be limited to a few people so that the device is not mishandled. Most decisions about the acquisition of diagnostic devices must take into account who will be able to use them, and here is the rub – if the department is small there may not be sufficient use of ultrasound to keep a user proficient, or if there is a high turnover of staff the amount of training required for new staff will be onerous. The ultrasound can be used in the investigation of many other illnesses, how will the device be retained for clinic use when it is in demand elsewhere? Who will do quality checks of scans and their interpretation?

How many machines are needed for what size of health centre and what is the best probe to buy that will be useful to the most children with different problems?

Will the use of CXR diminish or will LUS be added to what is done at present?

This study was done in 2018, it would be useful to return to the two health units and find out what is being done now? Was the use of ultrasound sustained and if not, why not? If LUS are still being performed are quality checks carried out, what do the stored pictures look like?

Response: We have written three other manuscripts (published and in press) discussing many of the important points you raised in this thoughtful review, but those key topics were beyond the scope of this present article.

Ginsburg AS et al. Performance of lung ultrasound in the diagnosis of pediatric pneumonia in Mozambique and Pakistan. *Pediatr Pulmonology* 2020. doi: 10.1002/ppul.25176. ahead of print
 Lung ultrasound patterns in pediatric pneumonia in Mozambique and Pakistan (in press)
 Serial lung ultrasounds in pediatric pneumonia in Mozambique and Pakistan (in press)

In conclusion this study raises more questions than it answers. The replacement of CXRs with LUSs is theoretically feasible and acceptable but the decision to do this must be context specific and constantly monitored.

Response: We acknowledge that these are important questions and should be the focus of further investigation. We did not address sustainability in this project, as LUS had not been implemented yet at the study sites nor is that currently planned. We appreciate that this study raised many questions (and more than it answered), and hope that others will read this work with the same level of forward thinking and thoughtfulness. We have attempted to address this focus in the final sentence of the Discussion: *“Future directions should include understanding barriers and facilitators to sustainability in LMIC following real-world experience with a particular focus on quality control, device maintenance and functionality, and acceptance and adoption of the new technology.”*

VERSION 2 – REVIEW

REVIEWER	Reza Rasti Karolinska Institutet, Sweden
REVIEW RETURNED	01-Feb-2021
GENERAL COMMENTS	Thank you for addressing all previous concerns.
REVIEWER	Elizabeth M Molyneux College of Medicine, Malawi
REVIEW RETURNED	01-Feb-2021
GENERAL COMMENTS	The authors have answer satisfactorily all the queries raised by the reviewers